# Learning Hierarchical Hyperbolic Embeddings for Compositional Zero-Shot Learning

## Abstract

Compositional zero-shot learning (CZSL) aims to recognize unseen state-object compositions by generalizing from a training set of their primitives (state and object). Current methods often overlook the rich hierarchical structures, such as the semantic hierarchy of primitives (*e.g.*, apple ⊏ fruit) and conceptual hierarchy between primitives and compositions (*e.g.*, sliced apple ⊏ apple). A few recent efforts have shown effectiveness in modeling these hierarchies through loss regularization within Euclidean space. In this paper, we argue that they fail to scale to the large-scale taxonomies required for real-world CZSL: the space's polynomial volume growth in *flat* geometry cannot match the exponential structure, impairing generalization capacity. To this end, we propose $H^2$EM, a new framework that learns Hierarchical Hyperbolic EMbeddings for CZSL. $H^2$EM leverages the unique properties of hyperbolic geometry, a space naturally suited for embedding tree-like structures with low distortion. However, a naive hyperbolic mapping may suffer from hierarchical collapse and poor fine-grained discrimination. We further design two learning objectives to structure this space: a taxonomic entailment loss that uses hyperbolic entailment cones to enforce the predefined hierarchies, and a discriminative alignment loss with hard negative mining to establish a large geodesic distance between semantically similar compositions. Extensive ablations on three benchmarks have demonstrated that $H^2$EM establishes a new state-of-the-art in both closed-world and open-world scenarios. Our codes will be released.

## 1 Introduction

How do we humans recognize novel concepts that have never been encountered before? This capability stems from generalizing learned knowledge to unseen domains (Mancini et al., 2021; 2022). Considering concepts like "sliced apple" and "ripe orange", we can recognize the unseen composition "ripe apple" by com-

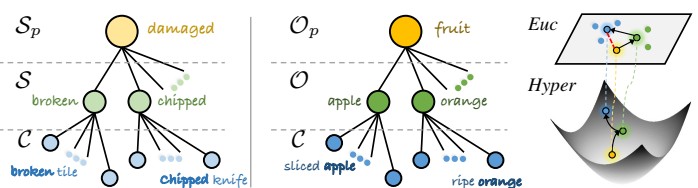

Figure 1: (**Left**) Hierarchical structures exist among primitives (states and objects) and compositions. (**Right**) The flat geometry of **Euc**lidean space distorts such tree-like structures, while the negative curvature of **Hyper**bolic space preserves the low-distortion embeddings.

bining the state "ripe" with the object "apple". Inspired by this cognitive process, Compositional Zero-Shot Learning (CZSL) tackles the challenge of identifying *unseen* state-object compositions during inference, leveraging visible primitives (*i.e.*, states and objects) from training concepts.

Recent work typically addresses CZSL by leveraging pre-trained large-scale vision-language models (VLMs), *e.g.*, CLIP (Radford et al., 2021). This paradigm measures similarities between a query image and a list of textual embeddings for each state, object, and composition (Nayak et al., 2023; Lu et al., 2023a). However, most of them ignore the rich hierarchical structures among categories: **i**) **Semantic Hierarchy**: Both state or object primitives are not semantically independent, but rather within larger semantic taxonomies (Wang et al., 2023). For instance, apple and orange are hyponyms of fruit (*cf.* Figure 1(Left)). This structure enables the model to infer common compositional affordances. By understanding that apple and orange are semantically related, a model can generalize its knowledge (*e.g.*, visual appearance of ripe) from seen ripe orange to rec-

ognize the unseen `ripe apple`. **ii**) **Conceptual Hierarchy**: An intrinsic entailment relationship exists where a composition is a more specific concept than its corresponding state or object. Take the composition `sliced apple` as an example, it is semantically entailed by the more general concepts of `sliced` and `apple`. Overlooking such a hierarchy damages the structural bond between a composition and its primitives, leaving the model unable to distinguish feasible unseen compositions from unfeasible ones, *e.g.*, `cloudy apple`, thus weakening the robustness (Ye et al., 2025).

Despite a few recent works beginning to explore these hierarchies (Wu et al., 2025; Ye et al., 2025), their efforts are fundamentally focused on the loss regularization (*e.g.*, logic constraints) within the common embedding space of VLMs. Admittedly, these can effectively model small, shallow hierarchical representations by pushing general concepts (*e.g.*, `fruit`) close to their many children, while keeping specific concepts (*e.g.*, `sliced apple`) only close to their immediate parents. Nevertheless, for CZSL's large-scale taxonomies, the *flat* geometry of VLM's Euclidean space creates a critical bottleneck. Its polynomial volume growth cannot accommodate the exponential structure of a large hierarchy without significant distortion (Murphy, 2012; Desai et al., 2023). This forces the numerous descendants of a general concept to be *crowded* into a small region, inevitably making embeddings of hierarchically distant concepts spatially close to each other (He et al., 2025a; Jun et al., 2025) (*cf.* red dashed line in Figure 1(Right)). Consequently, the model is prone to confusing distinct concepts, as the learned geometric distances cannot reflect the true hierarchical relationships.

In this paper, we propose $\mathbf{H}^2$**EM**, a framework that learns **H**ierarchical **H**yperbolic **EM**beddings for CZSL. $\mathbf{H}^2$**EM** projects features from a pre-trained Euclidean encoder into a hyperbolic manifold. Its exponential volume growth (Cannon et al., 1997) provides a natural geometric prior for embedding tree-like structures, where general concepts are central and specific concepts are peripheral,

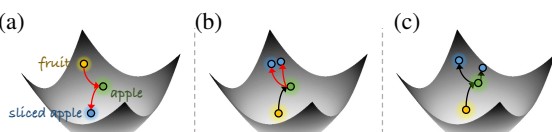

Figure 2: Challenges of direct Euc-to-Hyper transfer. (a) Specific concept and its general parent are in inverse hierarchy. (b) Semantically similar compositions are not sufficiently separated. (c) The desired hierarchical structure.

as shown in Figure 1(Right). On the other side, unlocking this potential for CZSL may face two challenges. **First**, it lacks explicit geometric constraints to enable models to leverage the two essential hierarchies mentioned. A naive mapping can lead to *hierarchical collapse*, where the embedding geometry inverts the true semantic hierarchy (*cf.* Figure 2a). To illustrate, the specific composition `sliced apple` might be mapped closer to the *origin* than its general parent `fruit`. **Second**, the commonly used contrastive objective is insufficient for fine-grained discrimination. By treating all negative classes equally, it fails to compel the model to establish a large geodesic distance between semantically similar compositions (*cf.* Figure 2b), such as `sliced apple` and `sliced orange`. Hence, their features may be close on the manifold, creating a less discriminative embedding space.

To this end, we devise two learning objectives to structure the hyperbolic manifold (*cf.* Figure 2c). For the **first** issue, we introduce a *taxonomic entailment loss*. This loss operates by enforcing geometric constraints, ensuring that child concepts (*e.g.*, `sliced apple`) lie within the entailment cone (Desai et al., 2023) of their more general parent concepts (*e.g.*, `sliced` and `apple`). To overcome the **second** challenge, we devise a *discriminative alignment loss* enhanced with hard negative mining. This objective allows the model to establish a large geodesic distance between semantically ambiguous compositions, thereby learning a more discriminative representation space.

To evaluate our $\mathbf{H}^2$EM, we conducted extensive experiments on three prevalent CZSL benchmarks: MIT-States (Isola et al., 2015), UT-Zappos (Naeem et al., 2021), and CGQA (Yu & Grauman, 2014). Experimental results show that $\mathbf{H}^2$EM establishes the new state-of-the-art in both closed-world and open-world scenarios, demonstrating its effectiveness and generalization capability. In summary, we made three main contributions in this paper: **i**) We are the first to explore the hierarchical embeddings in the hyperbolic space for CZSL. **ii**) We devise two hyperbolic learning objectives: a taxonomic entailment loss to enforce structural priors and a discriminative alignment loss with hard negative mining to learn fine-grained distinctions. **iii**) Extensive experiments on three benchmarks demonstrate $\mathbf{H}^2$EM's superiority with both qualitative and quantitative analysis.

## 2 RELATED WORK

**Compositional Zero-Shot Learning (CZSL).** CZSL addresses the challenge of recognizing novel state-object compositions at test time, given that each state or object is present in the training data.

Traditional CZSL methods can be divided into two main categories: 1) *Composed CZSL* (Anwaar et al., 2022; Mancini et al., 2022; Misra et al., 2017; Naeem et al., 2021; Purushwalkam et al., 2019), which seeks to classify compositional state-object pairs by mapping the holistic compositional visual features directly into a shared embedding space; and 2) *Decomposed CZSL* (Hao et al., 2023; Hu & Wang, 2023; Jiang & Zhang, 2024; Karthik et al., 2022; Khan et al., 2023; Li et al., 2022; 2020; Zhang et al., 2022), which separates the learning process by disentangling visual features for each primitive and employing distinct classifiers to independently predict states and objects. More recently, the advent of powerful VLMs, such as CLIP (Radford et al., 2021), has opened up new possibilities for leveraging large-scale pre-trained models to improve compositional generalization in zero-shot settings (Huang et al., 2024; Lu et al., 2023a;b; Nayak et al., 2023; Zheng et al., 2024). Building upon this VLM paradigm, this paper explores a new direction that explicitly models the hierarchical taxonomies by leveraging the exponential volume growth of hyperbolic space.

**Hyperbolic Representation Learning.** Hyperbolic geometry, characterized by its exponential volume growth with respect to radius, naturally lends itself to embedding data with tree-like or hierarchical structures. Due to these properties, hyperbolic representations have found wide applicability across various modalities, including text (Dhingra et al., 2018; Tifrea et al., 2016), images (Van Spengler et al., 2023; Atigh et al., 2022; Flaborea et al., 2024), graphs (Liu et al., 2019; Franco et al., 2023; Flaborea et al., 2024), and videos (Long et al., 2020). Recent developments have focused on learning hyperbolic embeddings via specialized neural network layers (Ganea et al., 2018a; Shimizu et al., 2021; He et al., 2025b; Chen et al., 2021; Lensink et al., 2022; Van Spengler et al., 2023), enhancing the capacity to capture hierarchical relationships. Notably, (Desai et al., 2023) introduces a hyperbolic entailment loss, which encourages child node embeddings to reside within the cone extended from their corresponding parent node embedding. This entailment loss has subsequently been leveraged for contrastive learning in VLMs to further improve representation quality for compositional and hierarchical data (Pal et al., 2025; Jun et al., 2025). This work moves beyond the cross-modal hierarchy of prior work to explicitly model the multi-level taxonomies that exist within each modality, encompassing both semantic and conceptual relationships.

## 3 METHODOLOGY

### 3.1 PRELIMINARIES

**Lorentz Model for Hyperbolic Geometry.** To effectively model the large tree-like taxonomies in compositional concepts, we operate in hyperbolic space, a Riemannian manifold with constant negative curvature $-\kappa$ ($\kappa > 0$). Following (Desai et al., 2023; Pal et al., 2025), we adopt the Lorentz model to model hyperbolic space due to its computational efficiency and well-defined neural operations (Cannon et al., 1997). Formally, the Lorentz model represents a $d$-dimensional hyperbolic manifold $\mathbb{L}^{d,\kappa}$ as the upper half of a two-sheeted hyperboloid in $(d + 1)$-dimensional Minkowski spacetime (Chen et al., 2021). Each point $\boldsymbol{p} \in \mathbb{L}^{d,\kappa}$ in this hyperbolic manifold is a vector $[p_0, \tilde{\boldsymbol{p}}]$ with a *time-like component* $p_0 \in \mathbb{R}$ and a *space-like component* $\tilde{\boldsymbol{p}} \in \mathbb{R}^d$, satisfying the condition $\langle \boldsymbol{p}, \boldsymbol{p} \rangle_{\mathbb{L}} = -1/\kappa$. Here, the Lorentzian inner product $\langle \cdot, \cdot \rangle_{\mathbb{L}}$ is formulated as:

$$\langle \boldsymbol{p}, \boldsymbol{q} \rangle_{\mathbb{L}} = -p_0 q_0 + \langle \tilde{\boldsymbol{p}}, \tilde{\boldsymbol{q}} \rangle_{\mathbb{E}}, \tag{1}$$

where $\langle \cdot, \cdot \rangle_{\mathbb{E}}$ denotes the Euclidean inner product. The geodesic distance (*i.e.*, the shortest path) between two points on the manifold is then calculated by this inner product:

$$d_{\mathbb{L}}(\boldsymbol{p}, \boldsymbol{q}) = \frac{1}{\sqrt{\kappa}} \text{arccosh}(-\kappa \langle \boldsymbol{p}, \boldsymbol{q} \rangle_{\mathbb{L}}). \tag{2}$$

Each point $\boldsymbol{p}$ on the manifold is associated with a $d$-dimensional tangent space $T_{\boldsymbol{p}}\mathbb{L}^{d,\kappa}$, which is a Euclidean space providing a local linear approximation. The exponential map (§3.2.2) is generally utilized to project a vector from this tangent space (*i.e.*, Euclidean space $T_{\boldsymbol{p}}\mathbb{L}^{d,\kappa}$) back onto the hyperboloid (*i.e.*, hyperbolic space $\mathbb{L}^{d,\kappa}$) (Khrulkov et al., 2020).

### 3.2 HIERARCHICAL HYPERBOLIC EMBEDDING BASED FRAMEWORK

In this section, we introduce our H$^2$EM framework, designed to explicitly model the rich hierarchical structures among compositional concepts, as illustrated in Figure 3. We first construct the tree-like taxonomy and then elaborate each component of our H$^2$EM framework.

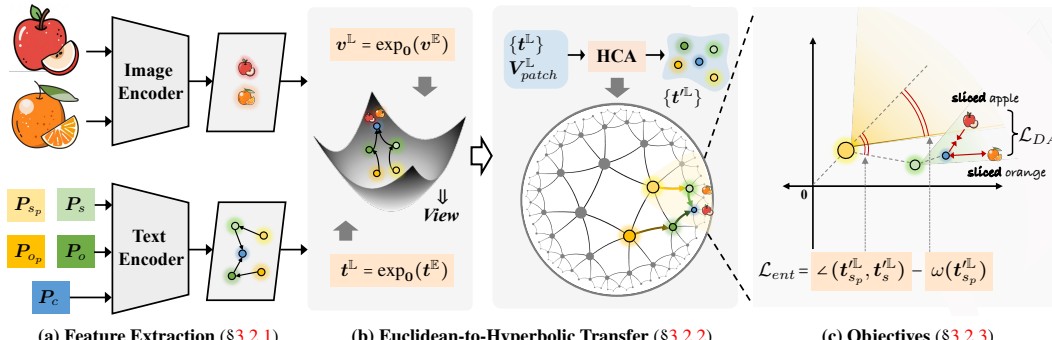

(a) Feature Extraction (§3.2.1)    (b) Euclidean-to-Hyperbolic Transfer (§3.2.2)    (c) Objectives (§3.2.3)

Figure 3: Overview of the $H^2$EM framework. (a) Extracting image and text features in Euclidean space. (b) Projecting features into hyperbolic space to model their hierarchy. (c) Jointly optimizing with a taxonomic entailment loss to enforce this hierarchy, and a discriminative alignment loss for cross-modal matching.

**Problem Formulation.** In CZSL, the goal is to classify an image $I$ into a composition category $c \in \mathcal{C}$. The category set $\mathcal{C}$ is formed by the Cartesian product of given state categories $\mathcal{S} = \{s_1, \ldots, s_{|\mathcal{S}|}\}$ and object categories $\mathcal{O} = \{o_1, \ldots, o_{|\mathcal{O}|}\}$, $i.e.$, $\mathcal{C} = \mathcal{S} \times \mathcal{O}$. The model is trained exclusively on a subset of *seen* composition and then evaluated under two protocols. **In closed-world testing**, the model must classify images from a fixed test set composed of both seen ($\mathcal{C}_{seen}$) and the unseen ($\mathcal{C}_{unseen}$) compositions, where $|\mathcal{C}_{seen} \cup \mathcal{C}_{unseen}| \ll |\mathcal{C}|$. **In open-world testing**, the model should generalize to any feasible composition (Nayak et al., 2023) within the full space $\mathcal{C}$.

**Tree-like Taxonomy Construction.** We construct a three-layer tree structure that encompasses two types of hierarchical relationships (§1). 1) **Semantic Hierarchy**. To explore the semantic hierarchy among the primitive concepts, we leverage Large Language Models (LLMs), *e.g.*, Gemini (Team et al., 2024), to automatically generate the hypernym (*i.e.*, the parent category) for each state and object primitive (More details are left in §C of the Appendix). The parent state and object categories are denoted as $\mathcal{S}_p$ and $\mathcal{O}_p$. 2) **Conceptual Hierarchy**. This inherent hierarchy defines the parent-child relationship between primitives and their compositions. To be specific, directed edges are created from each state $s \in \mathcal{S}$ and object $o \in \mathcal{O}$ to their corresponding composition $c = (s, o) \in \mathcal{C}$. These relationships can be formally denoted as the entailments $c \subset s$ and $c \subset o$. By integrating these two structures, we form the three-like taxonomy $\mathcal{T}$ that serves as a rich structural prior for training.

### 3.2.1 FEATURE EXTRACTION

**Visual Modality.** Given an input image $I \in \mathbb{R}^{H \times W \times 3}$, we employ the image encoder $E_v$ of CLIP (Radford et al., 2021) to extract visual features. The image encoder converts the input image into flattened patches ($N$ in total) along with a [CLS] token and positional embeddings, then generates a sequence of patch tokens. This sequence is processed by a series of self-attention blocks to generate visual embeddings $\boldsymbol{V}^{\mathbb{E}}_{patch} \in \mathbb{R}^{N+1,d}$. Finally, the output embedding corresponding to [CLS] token is projected by a linear layer to produce the final holistic image representation $\boldsymbol{v}^{\mathbb{E}} \in \mathbb{R}^d$.

To align with the primitives, we leverage the three-path paradigm to decompose the entangled features (Huang et al., 2024; Qu et al., 2025; Wu et al., 2025). Specifically, a state disentangler $D_s$, and an object disentangler $D_o$ are implemented as two Multi-Layer Perceptrons (MLPs) that extract primitive-specific features from the global representation. The three-path visual features of state, object, and composition in Euclidean space are as follows:

$$\boldsymbol{v}^{\mathbb{E}}_s = D_s(\boldsymbol{v}^{\mathbb{E}}), \quad \boldsymbol{v}^{\mathbb{E}}_o = D_o(\boldsymbol{v}^{\mathbb{E}}), \quad \boldsymbol{v}^{\mathbb{E}}_c = \boldsymbol{v}^{\mathbb{E}}. \tag{3}$$

**Textual Modality.** To capture the full semantic hierarchy, we generate text embeddings for three conceptual levels: compositions ($c \in \mathcal{C}$), primitives (states $s \in \mathcal{S}$ and objects $o \in \mathcal{O}$), and their abstract parent categories ($s_p, o_p$). Following (Nayak et al., 2023; Lu et al., 2023a; Huang et al., 2024), we employ the soft prompt tuning strategy. To elaborate, each prompt is a sequence of embeddings constructed from two components: a set of shared, learnable prefix context vectors and a learnable vocabulary token. For instance, the prompt representation for a state is a sequence $\boldsymbol{P}_s = [\boldsymbol{c}_1, ..., \boldsymbol{c}_l, \boldsymbol{s}]$, where $\{\boldsymbol{c}_1, ..., \boldsymbol{c}_l\}$ are the context vectors and $\boldsymbol{s}$ is the learnable token for state $s$. To capture the distinct semantic roles of states, objects, and compositions, the context vectors are

not shared across each branch (Huang et al., 2024). These prompt representations are then fed into the frozen CLIP's text encoder $E_t$ to obtain the corresponding Euclidean features, formulated as:

$$\boldsymbol{t}_{s_p}^{\mathbb{E}} = E_t(\boldsymbol{P}_{s_p}), \quad \boldsymbol{t}_{o_p}^{\mathbb{E}} = E_t(\boldsymbol{P}_{o_p}), \quad \boldsymbol{t}_s^{\mathbb{E}} = E_t(\boldsymbol{P}_s), \quad \boldsymbol{t}_o^{\mathbb{E}} = E_t(\boldsymbol{P}_o), \quad \boldsymbol{t}_c^{\mathbb{E}} = E_t(\boldsymbol{P}_c). \tag{4}$$

The extracted Euclidean visual and text features are subsequently projected into hyperbolic space.

### 3.2.2 EUCLIDEAN-TO-HYPERBOLIC TRANSFER

After feature extraction, we transfer the Euclidean representations into the hyperbolic manifold by first using the exponential map for projection, and then employing a hyperbolic cross-modal attention module to refine the text embeddings with image-specific visual context.

**Exponential Map.** To leverage the geometric property of the hyperbolic manifold, we map all Euclidean features into the hyperbolic embeddings of Lorentz model. This transition is achieved via the exponential map (Nickel & Kiela, 2018) at the origin $\boldsymbol{0}$, denoted as $\exp_{\boldsymbol{0}}: T_{\boldsymbol{0}}\mathbb{L}^{d,\kappa} \to \mathbb{L}^{d,\kappa}$. For ease of expression, $\boldsymbol{v}^{\mathbb{E}}$ is used to denote any Euclidean visual features, $e.g.$, $\boldsymbol{v}_s^{\mathbb{E}}$, $\boldsymbol{v}_o^{\mathbb{E}}$, $\boldsymbol{v}_c^{\mathbb{E}}$, and $\boldsymbol{V}_{patch}^{\mathbb{E}}$. Its corresponding hyperbolic embedding $\boldsymbol{v}^{\mathbb{L}} \in \mathbb{L}^{d,\kappa}$ is computed as:

$$\boldsymbol{v}^{\mathbb{L}} = \exp_{\boldsymbol{0}}(\boldsymbol{v}^{\mathbb{E}}) = \cosh(\sqrt{\kappa}\|\boldsymbol{v}^{\mathbb{E}}\|_{\mathbb{L}})\boldsymbol{0} + \frac{\sinh(\sqrt{\kappa}\|\boldsymbol{v}^{\mathbb{E}}\|_{\mathbb{L}})}{\sqrt{\kappa}\|\boldsymbol{v}^{\mathbb{E}}\|_{\mathbb{L}}}\boldsymbol{v}^{\mathbb{E}}, \tag{5}$$

where $\boldsymbol{0} = (\sqrt{1/\kappa}, 0, ..., 0)^T$, and the Lorentzian norm $\|\boldsymbol{v}^{\mathbb{E}}\|_{\mathbb{L}} = \sqrt{|\langle \boldsymbol{v}^{\mathbb{E}}, \boldsymbol{v}^{\mathbb{E}} \rangle_{\mathbb{L}}|}$. Similarly to the visual branch, we also utilize the exponential map to project all Euclidean text feature vectors into the Lorentz manifold $\mathbb{L}^{d,\kappa}$, denoted as $\boldsymbol{t}^{\mathbb{L}} = \exp_{\boldsymbol{0}}(\boldsymbol{t}^{\mathbb{E}})$.

**Hyperbolic Cross-Modal Attention.** Although soft-prompt tuning yields adaptive textual representations, the same semantic concept may not be optimal for aligning with the diverse visual content across all images (Huang et al., 2024), $e.g.$, the appearance of `sliced` on an `apple`'s surface versus on a `cake` is highly distinct. To this end, we devise a residual **Hyperbolic Cross-Modal Attention (HCA)** module to adaptively refine the text representations in the specific visual context:

$$\boldsymbol{t}'^{\mathbb{L}} = \boldsymbol{t}^{\mathbb{L}} \oplus_\kappa \lambda\text{HCA}(\boldsymbol{t}^{\mathbb{L}}, \boldsymbol{V}_{patch}^{\mathbb{L}}, \boldsymbol{V}_{patch}^{\mathbb{L}}), \tag{6}$$

where $\oplus_\kappa$ denotes Möbius addition (Ganea et al., 2018a), and $\lambda$ represents a learnable fusion parameter. $\boldsymbol{t}'^{\mathbb{L}}$ denotes any refined hyperbolic text embedding. The HCA module is defined as:

$$\text{HCA}(\boldsymbol{q}^{\mathbb{L}}, \boldsymbol{K}^{\mathbb{L}}, \boldsymbol{V}^{\mathbb{L}}) = \boldsymbol{q}^{\mathbb{L}} \oplus_\kappa \text{FFN}_{\mathbb{L}}(\text{MHA}_{\mathbb{L}}(\boldsymbol{q}^{\mathbb{L}}, \boldsymbol{K}^{\mathbb{L}}, \boldsymbol{V}^{\mathbb{L}})). \tag{7}$$

Here, the Feed-Forward Network $\text{FFN}_{\mathbb{L}}$ is composed of two hyperbolic linear layers (Yang et al., 2024) $\text{HFC}(\boldsymbol{x}) = [\sqrt{\|\boldsymbol{W}\boldsymbol{x} + \boldsymbol{b}\|^2 + 1/\kappa}, \boldsymbol{W}\boldsymbol{x} + \boldsymbol{b}]$, followed by a non-linear activation, where $\boldsymbol{W}$ and $\boldsymbol{b}$ denote the learnable weight and bias, respectively. The query $\boldsymbol{q}^{\mathbb{L}}$, key $\boldsymbol{K}^{\mathbb{L}}$, and value $\boldsymbol{K}^{\mathbb{L}}$ are derived through the hyperbolic linear layer with different weights, respectively. The multi-head attention $\text{MHA}_{\mathbb{L}}$ employs a hyperbolic linear attention mechanism (Yang et al., 2024) that first operates on the space-like components of the Lorentz vectors. The output for the $h$-th head's space-like components is computed by:

$$\tilde{\boldsymbol{Z}}_h = \frac{\phi(\tilde{\boldsymbol{Q}}_h)(\phi(\tilde{\boldsymbol{K}}_h)^T \tilde{\boldsymbol{V}}_h)}{\phi(\tilde{\boldsymbol{Q}}_h)(\phi(\tilde{\boldsymbol{K}}_h)^T \mathbf{1})}, \tag{8}$$

where $\mathbf{1}$ denotes an all-ones vector. $\phi(\cdot)$ is a non-linear feature map designed to enhance focus. It is defined as $\phi(\boldsymbol{x}) = \frac{\|\bar{\boldsymbol{x}}\|}{\|\bar{\boldsymbol{x}}^p\|}\bar{\boldsymbol{x}}^p$, where $\bar{\boldsymbol{x}} = \text{ReLU}(\boldsymbol{x})/t$ and $p$ is a power parameter and $t$ is a learnable scaling factor. The full hyperbolic output vectors $\boldsymbol{Z}_h^{\mathbb{L}} = [\boldsymbol{Z}_{h,0}, \tilde{\boldsymbol{Z}}_h]$ are then reconstructed by recalibrating their time-like components $\boldsymbol{Z}_{h,0} = \sqrt{\|\tilde{\boldsymbol{Z}}_h\|^2 + 1/\kappa}$. The outputs of all heads are concatenated and projected to form the final $\text{MHA}_{\mathbb{L}}$ output.

**Classification.** Since our $\text{H}^2\text{EM}$ relies on the Lorentz manifold's geometry, the state, object, and composition classification probabilities are based on the scaled negative geodesic distance $-d_{\mathbb{L}}$ between hyperbolic visual representation $\boldsymbol{v}^{\mathbb{L}}$ and refined text representation $\boldsymbol{t}'^{\mathbb{L}}$, written as:

$$p(s_i|I) = \frac{\exp(-d_{\mathbb{L}}(\boldsymbol{v}_s^{\mathbb{L}}, \boldsymbol{t}_{s_i}'^{\mathbb{L}})/\tau)}{\sum_{k=1}^{|\mathcal{S}|}\exp(-d_{\mathbb{L}}(\boldsymbol{v}_s^{\mathbb{L}}, \boldsymbol{t}_{s_k}'^{\mathbb{L}})/\tau)}, p(o_i|I) = \frac{\exp(-d_{\mathbb{L}}(\boldsymbol{v}_o^{\mathbb{L}}, \boldsymbol{t}_{o_i}'^{\mathbb{L}})/\tau)}{\sum_{k=1}^{|\mathcal{O}|}\exp(-d_{\mathbb{L}}(\boldsymbol{v}_o^{\mathbb{L}}, \boldsymbol{t}_{o_k}'^{\mathbb{L}})/\tau)}, p(c_i|I) = \frac{\exp(-d_{\mathbb{L}}(\boldsymbol{v}_c^{\mathbb{L}}, \boldsymbol{t}_{c_i}'^{\mathbb{L}})/\tau)}{\sum_{k=1}^{|\mathcal{C}|}\exp(-d_{\mathbb{L}}(\boldsymbol{v}_c^{\mathbb{L}}, \boldsymbol{t}_{c_k}'^{\mathbb{L}})/\tau)}, \tag{9}$$

where $\tau$ is the temperature parameter. During **inference**, we refine the compositional prediction by incorporating the primitive scores as assistance. As in (Huang et al., 2024; Wu et al., 2025; Qu et al., 2025), the final prediction $\hat{c}$ is given by:

$$\hat{c} = \arg\max_{c \in \mathcal{C}} p(c_{s,o}|I) + p(s|I) \cdot p(o|I). \tag{10}$$

### 3.2.3 TRAINING OBJECTIVES

The training objectives consist of three components: two main losses designed to structure the hyperbolic embedding space and learn discriminative features, and an extra primitive auxiliary loss.

**Taxonomic Entailment Loss.** To alleviate hierarchical collapse and enforce the predefined hierarchical structure of our taxonomy $\mathcal{T}$, we devise the *taxonomic entailment loss* $\mathcal{L}_{TE}$. As illustrated in Figure 3c, this loss leverages hyperbolic entailment cones (Ganea et al., 2018b; Desai et al., 2023), wherein a parent concept $\boldsymbol{p}$ geometrically entails its child concepts $\boldsymbol{q}$ by constraining them to lie within its cone-shaped region. We apply this loss to preserve both semantic hierarchy and conceptual hierarchy. For each parent-child edge $(\boldsymbol{p}, \boldsymbol{q}) \in \mathcal{T}$, the penalty is defined by the extent to which the child $\boldsymbol{q}$ is outside the parent's entailment cone:

$$\mathcal{L}_{ent}(\boldsymbol{p}, \boldsymbol{q}) = \max(0, \angle(\boldsymbol{p}, \boldsymbol{q}) - \omega(\boldsymbol{p})), \tag{11}$$

$$\angle(\boldsymbol{p}, \boldsymbol{q}) = \arccos\left(\frac{p_0 + q_0 \kappa \langle \boldsymbol{p}, \boldsymbol{q} \rangle_{\mathbb{L}}}{\|\tilde{\boldsymbol{q}}\| \sqrt{(\kappa \langle \boldsymbol{p}, \boldsymbol{q} \rangle_{\mathbb{L}})^2 - 1}}\right), \quad \omega(\boldsymbol{p}) = \arcsin\left(\frac{2\gamma}{\sqrt{\kappa} \|\tilde{\boldsymbol{p}}\|}\right), \tag{12}$$

where $\angle(\cdot, \cdot)$ is the exterior angle to the cone, $\omega(\cdot)$ is the cone's aperture. The constant $\gamma = 0.1$ sets boundary conditions near the origin. The penalty is zero if the child embedding is already inside the cone. Considering two hierarchical types in our taxonomy $\mathcal{T}$, whole $\mathcal{L}_{TE}$ is calculated by:

$$\mathcal{L}_{TE} = \mathcal{L}_{ent}(\boldsymbol{v}_s^{\mathbb{L}}, \boldsymbol{v}_c^{\mathbb{L}}) + \mathcal{L}_{ent}(\boldsymbol{v}_o^{\mathbb{L}}, \boldsymbol{v}_c^{\mathbb{L}})$$
$$+ \underbrace{\mathcal{L}_{ent}(\boldsymbol{t}_s'^{\mathbb{L}}, \boldsymbol{t}_c'^{\mathbb{L}}) + \mathcal{L}_{ent}(\boldsymbol{t}_o'^{\mathbb{L}}, \boldsymbol{t}_c'^{\mathbb{L}})}_{\text{Conceptual Hierarchy}} + \underbrace{\mathcal{L}_{ent}(\boldsymbol{t}_{s_p}'^{\mathbb{L}}, \boldsymbol{t}_s'^{\mathbb{L}}), + \mathcal{L}_{ent}(\boldsymbol{t}_{o_p}'^{\mathbb{L}}, \boldsymbol{t}_o'^{\mathbb{L}})}_{\text{Semantic Hierarchy}}. \tag{13}$$

**Discriminative Alignment Loss.** To enhance fine-grained discrimination, we devise the *discriminative alignment loss* $\mathcal{L}_{DA}$. In contrast to conventional contrastive loss that treats all negatives uniformly, our approach employs a hard negative mining strategy. For a positive sample with composition $(s_i, o_i)$, hard negatives are defined as compositions sharing the same primitive, *i.e.*, $\mathcal{H}_i = \{(s_j, o_k) \in \mathcal{C} \mid (s_j = s_i) \vee (o_k = o_i)\} \setminus \{(s_i, o_i)\}$. For instance, in Figure 3c, `sliced apple` and `sliced orange` serve as hard negatives for each other. The loss is a weighted InfoNCE-style (Oord et al., 2018) function that applies a larger penalty to these hard negatives:

$$\mathcal{L}_{DA} = -\log\left(\frac{\exp(-d_{\mathbb{L}}(\boldsymbol{v}_c^{\mathbb{L}}, \boldsymbol{t}_{c_i}'^{\mathbb{L}})/\tau)}{\sum_{c_k \in \mathcal{C} \setminus \mathcal{H}_i} \exp(-d_{\mathbb{L}}(\boldsymbol{v}_c^{\mathbb{L}}, \boldsymbol{t}_{c_k}'^{\mathbb{L}})/\tau) + w \sum_{c_j \in \mathcal{H}_i} \exp(-d_{\mathbb{L}}(\boldsymbol{v}_c^{\mathbb{L}}, \boldsymbol{t}_{c_j}'^{\mathbb{L}})/\tau)}\right), \tag{14}$$

where $w$ denotes the penalty weight of the hard negative samples.

**Primitive Auxiliary Loss.** In addition to the main compositional loss, we apply standard contrastive losses to the state and object branches to ensure that the disentangled visual features ($\boldsymbol{v}_s^{\mathbb{L}}$ and $\boldsymbol{v}_o^{\mathbb{L}}$) correctly align with their textual counterparts:

$$\mathcal{L}_s = -\log p(s|I), \quad \mathcal{L}_o = -\log p(o|I). \tag{15}$$

Finally, the total loss is a weighted sum with coefficients $\{\beta_i\}$ that integrates three objectives:

$$\mathcal{L}_{total} = \beta_1 \mathcal{L}_{DA} + \beta_2 \mathcal{L}_{TE} + \beta_3 (\mathcal{L}_s + \mathcal{L}_o). \tag{16}$$

## 4 EXPERIMENTS

### 4.1 EXPERIMENT SETTINGS

**Datasets.** We conducted experiments on three benchmarks: 1) **MIT-States** (Isola et al., 2015) contains 53k images (115 states, 245 objects). In the closed-world setting, the search space consists of 1,262 seen compositions and 300/400 unseen compositions for validation/testing. 2) **UT-Zappos** (Naeem et al., 2021) comprises 50k shoe images, covering 16 states and 12 objects. Following (Purushwalkam et al., 2019), closed-world setting adopts 83 seen and 15/18 unseen validation/testing compositions. 3) **CGQA** (Yu & Grauman, 2014) includes 39k images (453 states, 870 objects), with 5,592 seen for training and 1,040/923 unseen compositions for validation/testing. In the open-world settings, these datasets contain 28,175, 192, and 278,362 compositions, respectively.

**Metrics.** Following the established CZSL evaluation protocol (Mancini et al., 2021), we reported four metrics: 1) **Seen** evaluates the accuracy of seen compositions. 2) **Unseen** measures the accuracy

Table 1: Quantitative results (§4.2) on MIT-States (Isola et al., 2015), UT-Zappos (Yu & Grauman, 2014) and CGQA (Naeem et al., 2021) within *Closed-World* setting. ⋆ denotes reproduction results using official codes.

| Method | MIT-States | | | | UT-Zappos | | | | CGQA | | | |
|---|---|---|---|---|---|---|---|---|---|---|---|---|
| | Seen↑ | Unseen↑ | HM↑ | AUC↑ | Seen↑ | Unseen↑ | HM↑ | AUC↑ | Seen↑ | Unseen↑ | HM↑ | AUC↑ |
| CLIP (Radford et al., 2021)[ICML'21] | 30.2 | 46.0 | 26.1 | 11.0 | 15.8 | 49.1 | 15.6 | 5.0 | 7.5 | 25.0 | 8.6 | 1.4 |
| CoOp (Zhou et al., 2022)[IJCV'22] | 34.4 | 47.6 | 29.8 | 13.5 | 52.1 | 49.3 | 34.6 | 18.8 | 20.5 | 26.8 | 17.1 | 4.4 |
| Co-CGE (Mancini et al., 2022)[TPAMI'22] | 38.1 | 20.0 | 17.7 | 5.6 | 59.9 | 56.2 | 45.3 | 28.4 | 33.2 | 3.9 | 5.3 | 0.9 |
| ProDA (Lu et al., 2022)[CVPR'22] | 37.5 | 18.3 | 17.3 | 5.1 | 63.9 | 34.6 | 34.3 | 18.4 | - | - | - | - |
| CSP (Nayak et al., 2023)[ICLR'23] | 46.6 | 49.9 | 36.3 | 19.4 | 64.2 | 66.2 | 46.6 | 33.0 | 28.8 | 26.8 | 20.5 | 6.2 |
| DFSP(i2t) (Lu et al., 2023a)[CVPR'23] | 47.4 | 52.4 | 37.2 | 20.7 | 64.2 | 66.4 | 45.1 | 32.1 | 35.6 | 29.3 | 24.3 | 8.7 |
| DFSP(BiF) (Lu et al., 2023a)[CVPR'23] | 47.1 | 52.8 | 37.7 | 20.8 | 63.3 | 69.2 | 47.1 | 33.5 | 36.5 | 32.0 | 26.2 | 9.9 |
| DFSP(t2i) (Lu et al., 2023a)[CVPR'23] | 46.9 | 52.0 | 37.3 | 20.6 | 66.7 | 71.7 | 47.2 | 36.0 | 38.2 | 32.0 | 27.1 | 10.5 |
| GIPCOL (Xu et al., 2024)[WACV'24] | 48.5 | 49.6 | 36.6 | 19.9 | 65.0 | 68.5 | 48.8 | 36.2 | 31.9 | 28.4 | 22.5 | 7.1 |
| CDS-CZSL (Li et al., 2024)[CVPR'24] | 50.3 | 52.9 | 39.2 | 22.4 | 63.9 | 74.8 | 52.7 | 39.5 | 38.3 | 34.2 | 28.1 | 11.1 |
| CAILA⋆ (Li et al., 2024)[WACV'24] | 51.4 | 53.3 | 39.7 | 23.2 | 66.8 | 72.5 | 56.0 | 42.5 | 44.3 | 35.8 | 31.4 | 13.9 |
| Troika (Huang et al., 2024)[CVPR'24] | 49.0 | 53.0 | 39.3 | 22.1 | 66.8 | 73.8 | 54.6 | 41.7 | 41.0 | 35.7 | 29.4 | 12.4 |
| PLID (Bao et al., 2023)[ECCV'24] | 49.7 | 52.4 | 39.0 | 22.1 | 67.3 | 68.8 | 52.4 | 38.7 | 38.8 | 33.0 | 27.9 | 11.0 |
| LOGICZSL (Wu et al., 2025)[CVPR'25] | 50.8 | 53.9 | 40.5 | 23.4 | 69.6 | 74.9 | 57.8 | 45.8 | 44.4 | 39.4 | 33.3 | 15.3 |
| PLO (Li et al., 2025)[MM'25] | 51.6 | 53.7 | 40.2 | 23.4 | 70.3 | **75.8** | 55.3 | 43.6 | 44.7 | 38.1 | 33.0 | 14.9 |
| **H²EM (Ours)** | **52.4** | **54.1** | **41.3** | **24.2** | **70.4** | 74.5 | **59.8** | **46.6** | **45.0** | **40.4** | **33.9** | **16.0** |

Table 2: Quantitative results (§4.2) on MIT-States (Isola et al., 2015), UT-Zappos (Yu & Grauman, 2014) and CGQA (Naeem et al., 2021) within *Open-World* setting. ⋆ denotes reproduction results using official codes.

| Method | MIT-States | | | | UT-Zappos | | | | CGQA | | | |
|---|---|---|---|---|---|---|---|---|---|---|---|---|
| | Seen↑ | Unseen↑ | HM↑ | AUC↑ | Seen↑ | Unseen↑ | HM↑ | AUC↑ | Seen↑ | Unseen↑ | HM↑ | AUC↑ |
| CLIP (Radford et al., 2021)[ICML'21] | 30.1 | 14.3 | 12.8 | 3.0 | 15.7 | 20.6 | 11.2 | 2.2 | 7.5 | 4.6 | 4.0 | 0.3 |
| CoOp (Zhou et al., 2022)[IJCV'22] | 34.6 | 9.3 | 12.3 | 2.8 | 52.1 | 31.5 | 28.9 | 13.2 | 21.0 | 4.6 | 5.5 | 0.7 |
| Co-CGE (Mancini et al., 2022)[TPAMI'22] | 38.1 | 20.0 | 17.7 | 5.6 | 59.9 | 56.2 | 45.3 | 28.4 | 33.2 | 3.9 | 5.3 | 0.9 |
| ProDA (Lu et al., 2022)[CVPR'22] | 37.5 | 18.3 | 17.3 | 5.1 | 63.9 | 34.6 | 34.3 | 18.4 | - | - | - | - |
| CSP (Nayak et al., 2023)[ICLR'23] | 46.3 | 15.7 | 17.4 | 5.7 | 64.1 | 44.1 | 38.9 | 22.7 | 28.7 | 5.2 | 6.9 | 1.2 |
| DFSP(i2t) (Lu et al., 2023a)[CVPR'23] | 47.4 | 52.4 | 37.2 | 20.7 | 64.2 | 66.4 | 45.1 | 32.1 | 35.6 | 29.3 | 24.3 | 8.7 |
| DFSP(BiF) (Lu et al., 2023a)[CVPR'23] | 47.1 | 18.1 | 19.2 | 6.7 | 63.5 | 57.2 | 42.7 | 27.6 | 36.4 | 7.6 | 10.6 | 2.4 |
| DFSP(t2i) (Lu et al., 2023a)[CVPR'23] | 47.5 | 18.5 | 19.3 | 6.8 | 66.8 | 60.0 | 44.0 | 30.3 | 38.3 | 7.2 | 10.4 | 2.4 |
| GIPCOL (Xu et al., 2024)[WACV'24] | 48.5 | 16.0 | 17.9 | 6.3 | 65.0 | 45.0 | 40.1 | 23.5 | 31.6 | 5.5 | 7.3 | 1.3 |
| CDS-CZSL (Li et al., 2024)[CVPR'24] | 49.4 | 21.8 | 22.1 | 8.5 | 64.7 | 61.3 | 48.2 | 32.3 | 37.6 | 8.2 | 11.6 | 2.7 |
| CAILA⋆ (Li et al., 2024)[WACV'24] | 51.4 | 20.1 | 20.9 | 8.0 | 65.1 | 59.6 | 44.8 | 29.9 | 43.8 | 11.4 | 7.9 | 3.3 |
| Troika (Huang et al., 2024)[CVPR'24] | 48.8 | 18.7 | 20.1 | 7.2 | 66.4 | 61.2 | 47.8 | 33.0 | 40.8 | 7.9 | 10.9 | 2.7 |
| PLID (Bao et al., 2023)[ECCV'24] | 49.1 | 18.7 | 20.0 | 7.3 | 67.6 | 55.5 | 46.6 | 30.8 | 39.1 | 7.5 | 10.6 | 2.5 |
| LOGICZSL (Wu et al., 2025)[CVPR'25] | 50.7 | 21.4 | 22.4 | 8.7 | 69.6 | **63.7** | 50.8 | 36.2 | 43.7 | 9.3 | 12.6 | 3.4 |
| PLO (Li et al., 2025)[MM'25] | 49.7 | 19.4 | 21.4 | 7.8 | 68.0 | 63.5 | 47.8 | 33.1 | 43.9 | 10.4 | 13.9 | 3.9 |
| **H²EM (Ours)** | **52.4** | **22.2** | **23.3** | **9.4** | **70.4** | 62.3 | **54.5** | **38.8** | **45.1** | 10.8 | **15.2** | **4.4** |

exclusively for unseen compositions. 3) **Harmonic Mean (HM)** demotes the harmonic mean of the seen and unseen accuracy, serving as a comprehensive metric. 4) **Area Under the Curve (AUC)** is computed by integrating the area beneath the seen-unseen accuracy curve, which is a valuable indicator of a model's performance across a wide range of operating points from $-\infty$ to $+\infty$.

**Implementation Details.** Due to space constraints, details are provided in the Appendix (§A).

### 4.2 COMPARISON WITH THE STATE-OF-THE-ARTS

For a fair comparison, we evaluated H²EM against state-of-the-art CLIP-based methods that use the same ViT-L/14 backbone (Radford et al., 2021) in both closed-world and open-world settings.

**Quantitative Analysis.** The results are shown in Table 1 and Table 2. In the *closed-world setting*, H²EM consistently establishes new state-of-the-art results, surpassing the leading competitor LOGICZSL (Wu et al., 2025) across all datasets. Concretely, for the comprehensive metric HM, H²EM obtains **41.3**%, **59.8**%, and **33.9**% on MIT-States, UT-Zappos, and CGQA, separately. In the *open-world setting*: H²EM continues to demonstrate its superiority in this more difficult setting. It achieves the highest HM on MIT-States (**23.3**%) and the highest AUC on UT-Zappos (**54.5**%). Notably, for the most challenging CGQA dataset, our H²EM achieves the best performance across all metrics. Compared with LOGICZSL, our H²EM obtains significant gains, with improvements of **+1.4**% on seen compositions, **+1.5**% on unseen compositions, **+2.6**% on HM, and **+1.0**% on AUC. These consistency improvements demonstrate the effectiveness and robustness of our H²EM and its advanced hierarchical hyperbolic embeddings for compositional visual reasoning.

Table 3: Analysis of key components (§4.3) on MIT-States (Isola et al., 2015) and CGQA (Naeem et al., 2021) within **Closed-World** setting. Troika (Huang et al., 2024) serves as the baseline model.

| Euc⇒Hyper (§3.2.2) | $\mathcal{L}_{TE}$ (Eq.13) | $\mathcal{L}_{DA}$ (Eq.14) | MIT-States | | | | CGQA | | | |
|---|---|---|---|---|---|---|---|---|---|---|
| | | | Seen↑ | Unseen↑ | HM↑ | AUC↑ | Seen↑ | Unseen↑ | HM↑ | AUC↑ |
| ✗ | ✗ | ✗ | 49.0 | 53.0 | 39.3 | 22.1 | 41.0 | 35.7 | 29.4 | 12.4 |
| ✓ | ✗ | ✗ | 52.2 | 53.2 | 40.4 | 23.6 | 44.7 | 39.1 | 33.0 | 15.2 |
| ✓ | ✓ | ✗ | 52.1 | 53.8 | 41.2 | 24.0 | 44.9 | 39.5 | 33.8 | 15.6 |
| ✓ | ✗ | ✓ | 52.3 | 53.9 | 40.8 | 24.1 | **45.4** | 39.3 | 33.8 | 15.7 |
| ✓ | ✓ | ✓ | **52.4** | **54.1** | **41.3** | **24.2** | 45.0 | **40.4** | **33.9** | **16.0** |

Table 4: Ablation study (§4.3) on the coefficients $\beta_1$ and $\beta_2$ used in training objective Eq.16.

| Coefficient $\beta_1$ | MIT-States | | | |
|---|---|---|---|---|
| | Seen↑ | Uneen↑ | HM↑ | AUC↑ |
| $\beta_1 = 0.1$ | 45.1 | 51.0 | 36.6 | 19.4 |
| $\beta_1 = 0.5$ | 51.3 | 53.5 | 40.1 | 23.4 |
| $\beta_1 = 1.0$ | 52.4 | **54.1** | **41.3** | **24.2** |
| $\beta_1 = 1.5$ | **52.5** | 53.6 | 40.8 | 23.9 |
| $\beta_1 = 2.0$ | 52.0 | 53.9 | 40.6 | 23.8 |

(a) Discriminative Alignment Loss Coefficient $\beta_1$

| Coefficient $\beta_2$ | MIT-States | | | |
|---|---|---|---|---|
| | Seen↑ | Uneen↑ | HM↑ | AUC↑ |
| $\beta_2 = 0.01$ | 52.3 | 54.1 | 40.9 | 24.1 |
| $\beta_2 = 0.05$ | **52.6** | 54.0 | 41.1 | 24.2 |
| $\beta_2 = 0.1$ | 52.4 | **54.1** | **41.3** | **24.2** |
| $\beta_2 = 0.5$ | 51.5 | 52.3 | 39.4 | 22.2 |
| $\beta_2 = 1.0$ | 48.2 | 51.2 | 37.0 | 20.5 |

(b) Taxonomic Entailment Loss Coefficient $\beta_2$

## 4.3 ABLATION STUDY

**Key Component Analysis.** Contributions of hyperbolic embedding, taxonomic entailment loss ($\mathcal{L}_{TE}$), and discriminative alignment loss ($\mathcal{L}_{DA}$) were evaluated, as summarized in Table 3. The first row refers to the baseline model (*i.e.*, Troika (Huang et al., 2024) with both three-path paradigm and cross-modal refinement) implemented in Euclidean space. Three crucial conclusions can be drawn. **First**, by simply projecting features into a hyperbolic embedding space, significant and consistent improvements are observed, with **+1.1**% to **+3.6**% gains on the HM. This demonstrates that the inherent geometric properties of hyperbolic space are better suited for CZSL. **Second**, with the guidance of the taxonomic entailment loss, the model is constrained by the predefined hierarchy, resulting in further performance gains (*e.g.*, **+0.8**% HM on MIT-States). **Third**, benefiting from the discriminative alignment loss, the model is compelled to distinguish between semantically similar compositions. This yields notable improvements, particularly on the seen metric for CGQA (**+1.1**% compared to the base hyperbolic model). Combining all components allows for the best overall performance across all evaluation metrics, confirming their contributions to our H$^2$EM.

**Discriminative Alignment Loss Coefficient** $\beta_1$. We investigated the impact of the coefficient $\beta_1$, which controls the weight of discriminative alignment loss (Eq. 14). As shown in Table 4a, the model's performance peaks at $\beta_1 = 1.0$, achieving the best HM of 41.3% and AUC of 24.2%. Decreasing this weight significantly degrades performance, as the model lacks a strong discriminative signal. A weight greater than 1.0 also leads to a slight decline, likely by overpowering the other regularizing terms. Therefore, we set $\beta_1 = 1.0$ in our final model.

**Taxonomic Entailment Loss Coefficient** $\beta_2$. We analyzed the sensitivity of taxonomic entailment loss (Eq. 11). This loss acts as a geometric regularizer, infusing the predefined hierarchy into the embedding space. Table 4b shows that a small coefficient is optimal, with the best result achieved at $\beta_2 = 0.1$. While a smaller weight is still beneficial, a large weight (*e.g.*, $\beta_2 \geq 0.5$) causes a sharp drop in performance. This indicates that while the hierarchical constraint is crucial, an overly strong geometric prior can over-constrain the model and impede the primary classification objectives. Thus, we set $\beta_2 = 0.1$ to get a trade-off between geometric structure and discriminative learning.

## 4.4 QUALITATIVE ANALYSIS

**Visualization of Top-1 Predictions.** We visualized top-1 predictions in Figure 4. For **success cases**, our H$^2$EM model proves more robust than the Troika baseline, accurately recognizing compositions where the baseline fails. For instance, it correctly identifies `thin ring` while Troika incorrectly predicts `engraved plastic`. This demonstrates that by leveraging hyperbolic geometry, H²EM learns a semantic space that can distinguish visually similar but conceptually distinct concepts. As for **failure cases**, unlike the baseline, H$^2$EM's errors are often semantically feasible alternatives. A clear example is for the `old computer` image, where our model predicts the reasonable synonym

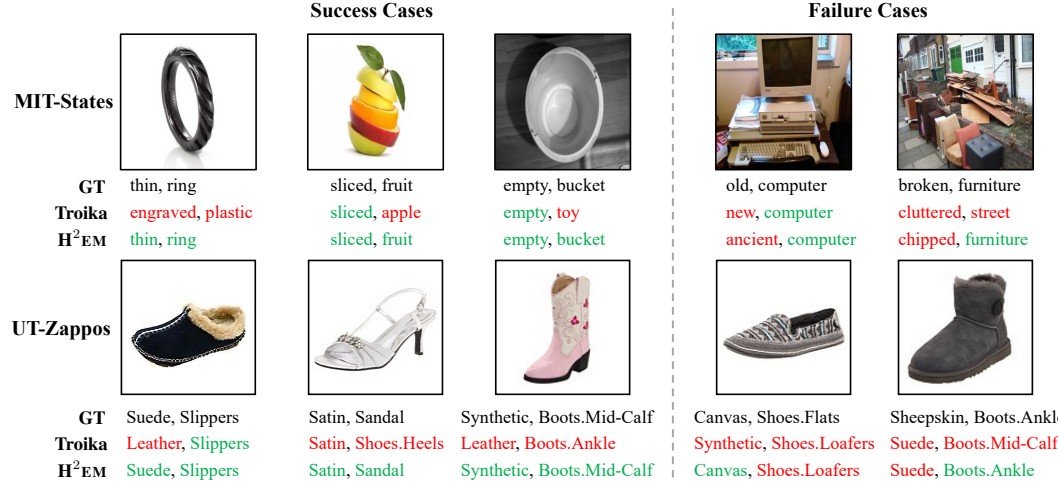

Figure 4: Visualization of Top-1 Predictions (§4.4). The comparison between our H²EM and Troika (Huang et al., 2024) on MIT-States (Isola et al., 2015) and UT-Zappos (Yu & Grauman, 2014). Correct and wrong predictions are marked in **green** and **red**, respectively.

`ancient`, while Troika predicts the antonym `new`. This indicates that our hierarchically structured space ensures that even incorrect predictions are semantically reasonable to the ground truth, leading to more logically sound failures.

**Visualization of Hyperbolic Space.** In Figure 5, we visualized the learned hyperbolic space for text embeddings using samples from MIT-States. We analyzed the norm distribution (Desai et al., 2023) of 2K samples and utilized 200 samples for the HoroPCA (Chami et al., 2021) visualization. Two key observations can be made. First, the norm distributions in Figure 5a reveal a distinct ordering: parent concepts are positioned closest to the origin,

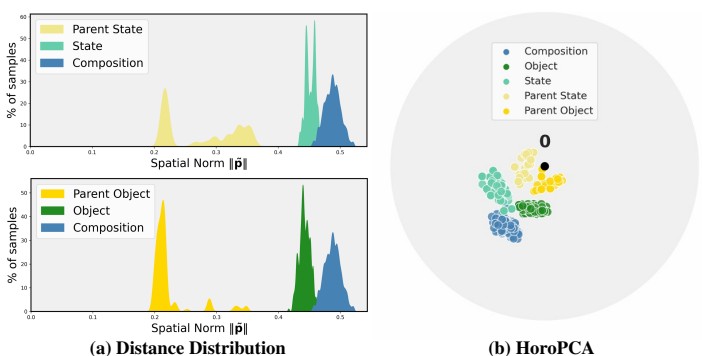

(a) Distance Distribution (b) HoroPCA

Figure 5: Visual results (§4.4) of text embeddings in learned hyperbolic space using samples from MIT-States (Isola et al., 2015).

followed by primitives, with compositions being the most peripheral. Second, the HoroPCA visualization in Figure 5b qualitatively confirms this, showing that the embeddings form distinct, concentric clusters corresponding to their level in the hierarchy. This clear hierarchical organization can be attributed to our taxonomic entailment loss (Eq. 11), which geometrically constrains the representations to adhere to the predefined parent-child relationships. Overall, these visualizations provide strong evidence that H²EM effectively embeds the symbolic taxonomy into the geometry of the hyperbolic space. A similar analysis for the visual modality is provided in the Appendix (§D.2).

## 5 CONCLUSION

In this work, we tackled the inadequacy of Euclidean-based models to represent the rich semantic hierarchy and conceptual hierarchy in compositional concepts. We introduced H²EM to learn hierarchical embeddings within hyperbolic space, a geometry naturally suited for CZSL's large tree-like taxonomy. Our H²EM prevents the hierarchical collapse and poor fine-grained discrimination by employing two tailored objectives: a taxonomic entailment loss to geometrically instill the symbolic hierarchy, and a discriminative alignment loss with hard negative mining to separate similar compositions. Extensive experiments on three gold-standard datasets demonstrate the effectiveness of H²EM in both closed-world and open-world scenarios. Future work can extend this hyperbolic framework to other compositional tasks or explore methods for automatic taxonomy discovery.

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

APPENDIX

This appendix is organized as follows:

- §A elaborates the implementation details of $H^2$EM.
- §B displays more experimental results.
- §C shows the prompt for semantic hierarchy construction.
- §D offers more qualitative results.
- §E provides the reproducibility statement.
- §F states the usage of large language models.

## A  IMPLEMENTATION DETAILS

### A.1  ARCHITECTURE

For the CLIP model, we utilized the official OpenAI checkpoints, opting for the Vision Transformer with a base configuration of ViT-L/14 (Radford et al., 2021). All visual and textual features were projected into a 768-dimensional embedding space. For the hyperbolic manifold, we used the Lorentz model (§3.1) with a learnable curvature $\kappa$, initialized at 1.0. Consistent with (Lu et al., 2023a; Huang et al., 2024), the learnable context length $l$ for the soft prompts was set to 8. For the hyperbolic linear layer and hyperbolic attention mechanism, we employed the same settings as (Yang et al., 2024). The state and object disentanglers ($D_s, D_o$, Eq. 3) were implemented as 2-layer MLPs with a RELU activation function, a standard choice established in (Huang et al., 2024; Wu et al., 2025). For fair comparisons, our HCA module (Eq. 7) adopted a 12-head attention mechanism, following (Huang et al., 2024). For the semantic hierarchy construction, the Gemini-2.5Pro (Comanici et al., 2025) model was employed as the LLM with the prompts in §C.

### A.2  TRAINING AND INFERENCE DETAILS

Our models were trained and evaluated on the NVIDIA H20 GPU using the PyTorch (Paszke et al., 2019) framework. Following the same configuration as (Lu et al., 2023a), we used the Adam optimizer (Loshchilov & Hutter, 2017) for all datasets, with a learning rate of 1e-4 and a batch size of 64 as default. For our contrastive loss functions (Eq. 9), the temperature $\tau$ was set to the same value as pretrained CLIP's logits scale. The hard negative weight $w$ in discriminative alignment loss (Eq. 14) was set to 3.0. Based on our ablation studies, the final loss coefficients (Eq. 16) were set to $\beta_1 = 1.0$, $\beta_2 = 0.1$, and $\beta_3 = 0.5$. In the open-world evaluation, we adhered to the post-training calibration method (Nayak et al., 2023) to filter out compositions deemed infeasible.

## B  MORE EXPERIMENTAL RESULTS

### B.1  ABLATION STUDY

**Hyperbolic Cross-Modal Attention.** We evaluated the effectiveness of our Hyperbolic Cross-Modal Attention (HCA) module (§3.2.2) in Table S1. In the *closed-world setting*, incorporating HCA yields significant improvements across all metrics, boosting the HM by **+2.0**% (from 39.3% to 41.3%) and the AUC by **+1.9**%. This confirms that adaptively refining the static text embeddings with instance-specific visual context is crucial for accurate alignment. As for the *open-world setting*, HCA still brings consistent improvements across all metrics. It enhances the HM by **+1.6**% and the AUC by **+1.3**%. Although the performance gains are slightly less pronounced than in the closed-world scenario, this demonstrates that instance-specific refinement is still beneficial for distinguishing correct compositions from the vast search space in the open world.

**Hard Negative Weight $w$.** We analyzed the impact of the hard negative weight $w$ in discriminative alignment loss (Eq. 14), with results in Table S2. A setting of $w = 1.0$ denotes a standard contrastive loss without weighting. Increasing the weight to $w = 3.0$ achieves the optimal performance, improving the HM from 41.2% to 41.3% and the AUC from 24.0% to 24.2%. This demonstrates that

Table S1: Ablation study (§B) of the Hyperbolic Cross-Modal Attention (HCA, §3.2.2) on MIT-States (Isola et al., 2015) under closed-world and open world settings.

| Method | Setting | MIT-States | | | |
|---|---|---|---|---|---|
| | | Seen↑ | Unseen↑ | HM↑ | AUC↑ |
| **Ours** | Closed-World | **52.4** | **54.1** | **41.3** | **24.2** |
| *w/o* HCA | | 49.8 | 52.3 | 39.3 | 22.3 |
| **Ours** | Open-World | **52.4** | **22.2** | **23.3** | **9.4** |
| *w/o* HCA | | 49.9 | 20.0 | 21.7 | 8.1 |

Table S2: Ablation study (§B) of the weight $w$ of hard negative samples (Eq. 14) on MIT-States.

| $w$ | MIT-States | | | |
|---|---|---|---|---|
| | Seen↑ | Unseen↑ | HM↑ | AUC↑ |
| 1.0 | 52.1 | 53.8 | 41.2 | 24.0 |
| 3.0 | **52.4** | **54.1** | **41.3** | **24.2** |
| 5.0 | 51.9 | 54.0 | 40.9 | 23.9 |
| 7.0 | 51.9 | 53.7 | 40.7 | 23.7 |
| 10.0 | 51.7 | 53.6 | 40.5 | 23.6 |

Table S3: Ablation study (§B) of the primitive auxiliary loss coefficient $\beta_3$ (Eq. 16) on MIT-States.

| $\beta_3$ | MIT-States | | | |
|---|---|---|---|---|
| | Seen↑ | Unseen↑ | HM↑ | AUC↑ |
| 0.1 | 51.9 | 53.6 | 40.5 | 23.6 |
| 0.3 | 52.1 | 54.0 | 41.0 | 24.0 |
| 0.5 | **52.4** | **54.1** | **41.3** | **24.2** |
| 0.7 | 52.4 | 53.9 | 40.8 | 24.0 |
| 1.0 | 52.2 | 53.6 | 40.7 | 23.8 |

Table S4: Ablation study (§B) the different backbone on MIT-States (Isola et al., 2015) under closed-world settings. We reported the standard error with 5-times experiments.

| Method | Backbone | MIT-States | | | |
|---|---|---|---|---|---|
| | | Seen↑ | Unseen↑ | HM↑ | AUC↑ |
| DFSP (Lu et al., 2023a) | | 36.7 | 43.4 | 29.4 | 13.2 |
| Troika (Huang et al., 2024) | ViT-B/32 | 39.5 | 42.8 | 30.5 | 13.9 |
| H$^2$EM | | **43.1**$_{\pm 0.4}$ | **45.4**$_{\pm 0.5}$ | **32.9**$_{\pm 0.3}$ | **16.2**$_{\pm 0.2}$ |
| DFSP (Lu et al., 2023a) | | 39.6 | 46.5 | 31.5 | 15.1 |
| Troika (Huang et al., 2024) | ViT-B/16 | 45.3 | 46.4 | 33.9 | 17.2 |
| H$^2$EM | | **46.9**$_{\pm 0.5}$ | **48.3**$_{\pm 0.4}$ | **35.8**$_{\pm 0.2}$ | **18.9**$_{\pm 0.1}$ |
| DFSP (Lu et al., 2023a) | | 46.8 | 52.2 | 37.4 | 20.6 |
| Troika (Huang et al., 2024) | ViT-L/14 | 49.0 | 53.0 | 39.3 | 22.1 |
| H$^2$EM | | **52.4**$_{\pm 0.5}$ | **54.1**$_{\pm 0.2}$ | **41.3**$_{\pm 0.4}$ | **24.2**$_{\pm 0.3}$ |

applying a moderate extra penalty on semantically similar compositions is beneficial, compelling the model to learn more discriminative features. However, increasing the weight further ($w \geq 5.0$) leads to a decline in performance, indicating that an excessive focus on the hardest negatives can destabilize training and harm overall generalization. Therefore, we set $w = 3.0$ in our experiments.

**Primitive Auxiliary Loss Coefficient** $\beta_3$. Table S3 shows our analysis of the coefficient $\beta_3$, which scales the auxiliary losses on the primitive state and object branches. This objective is important for regularizing the feature disentanglement process. The model's performance peaks at $\beta_3 = 0.5$, achieving the best HM of 41.3% and AUC of 24.2%. A smaller weight provides insufficient regularization for the disentangled features, while a larger weight ($\beta_3 \geq 0.7$) begins to detract from the main compositional learning objective, causing performance to decline. We thus set $\beta_3 = 0.5$ to best balance the learning of primitive and compositional features.

**Backbone Study.** To evaluate the robustness and generalizability of our approach, we conducted a backbone study on MIT-States, comparing H$^2$EM against strong baselines, *i.e.*, DFSP (Lu et al., 2023a) and Troika (Huang et al., 2024) across three different Vision Transformer (ViT) architectures. The results are reported in Table S4. The analysis shows two clear trends. First, our H$^2$EM consistently and significantly outperforms both DFSP and Troika across all tested backbones, from the smaller ViT-B/32 to the larger ViT-L/14. Notably, H$^2$EM maintains a stable and significant margin over Troika in HM, with gains ranging from +1.9% on ViT-B/16 to +2.4% on ViT-B/32. Second, as expected, the performance of all methods scales with the capacity of the backbone, with our model effectively leveraging the stronger features from ViT-L/14 to achieve its best results. The consistent superiority of H$^2$EM regardless of the feature extractor's strength demonstrates that our proposed hierarchical hyperbolic embeddings provide a robust improvement to the compositional reasoning.

Table S5: Efficiency comparison (§B.2) on UT-Zappos (Yu & Grauman, 2014). We report the number of trainable parameters, training time per epoch, and inference time of the whole test dataset.

| Method | # Params↓ | Training Time↓ | Inference Time↓ | UT-Zappos | | | |
| --- | --- | --- | --- | --- | --- | --- | --- |
| | | | | Seen↑ | Unseen↑ | HM↑ | AUC↑ |
| Troika (Huang et al., 2024) | 21.7M | 3.3min | 22s | 66.8 | 73.8 | 54.6 | 41.7 |
| **H$^2$EM (Ours)** | 21.8M | 3.4min | 24s | 70.4 | 74.5 | 59.8 | 46.6 |

---

**Question:** You are an expert linguist and knowledge graph constructor specializing in semantic hierarchies. Your task is to identify the general parent concept (hypernym) for each term in a given list of objects: {**category list**}.
**Rules:**
1. The parent concept must be a single, common English word. Do not use phrases.
2. Similar concepts must be mapped to the same parent for consistency.
3. The KEYS of the output object should be the general parent concepts you identify.
4. The VALUE for each key should be a list of all the child terms from the input that belong to that parent.
**Examples:**
   **fruit**: [**apple**, **banana**, **orange**, ...]
   **animal**: [**dog**, **cat**, **horse**, ...]

Figure S1: Semantic hierarchy construction prompt for object primitive.

## B.2 EFFICIENCY ANALYSIS

We conducted an efficiency analysis comparing our H$^2$EM with the baseline Troika (Huang et al., 2024), with results reported in Table S5. Our method demonstrates high efficiency, with a total of 21.8M trainable parameters, *i.e.*, a marginal increase of only 0.1M over Troika. This minor overhead is due to the learnable embeddings for the parent categories and the lightweight parameters of hyperbolic space learning. Consequently, the training and inference times are also negligibly impacted. In stark contrast to this minimal additional cost, our framework yields a substantial performance leap on UT-Zappos, achieving a +5.2% improvement in Harmonic Mean and a +4.9 gain in AUC. This analysis confirms that H$^2$EM achieves a significantly better performance-efficiency trade-off, underscoring its practical value.

## C SEMANTIC HIERARCHY CONSTRUCTION PROMPT

As mentioned in our main methodology (§3), we leverage LLMs to automatically construct the semantic hierarchy for our primitive concepts. Taking the object primitive for example, the prompt designed to guide the LLM for hierarchy construction is shown in Figure S1. The prompt comprises several key components: 1) a *role-playing directive* that instructs the model to act as an expert linguist; 2) a clear *task definition* to group primitives under parent concepts (hypernyms); 3) a set of explicit *rules* governing the output format and properties of the parent concepts; and 4) illustrative *in-context examples* to demonstrate the desired grouping. This similar prompt was also used to generate the hierarchy for state primitives.

## D MORE QUALITATIVE RESULTS

### D.1 VISUALIZATION OF TOP-1 PREDICTIONS

To further validate our framework, Figure S2 provides additional qualitative comparisons between H$^2$EM and the Troika baseline on the challenging CGQA dataset. In success cases, H$^2$EM is consistently more accurate. For instance, it correctly identifies `rood pizza` and `tall grass`, where Troika makes significant errors, including predicting the completely wrong `wood table` and direct antonym `short grass`. H$^2$EM also correctly focuses on the `white shirt` composition while the baseline is disturbed by other salient scene elements. The failure cases further highlight H$^2$EM's

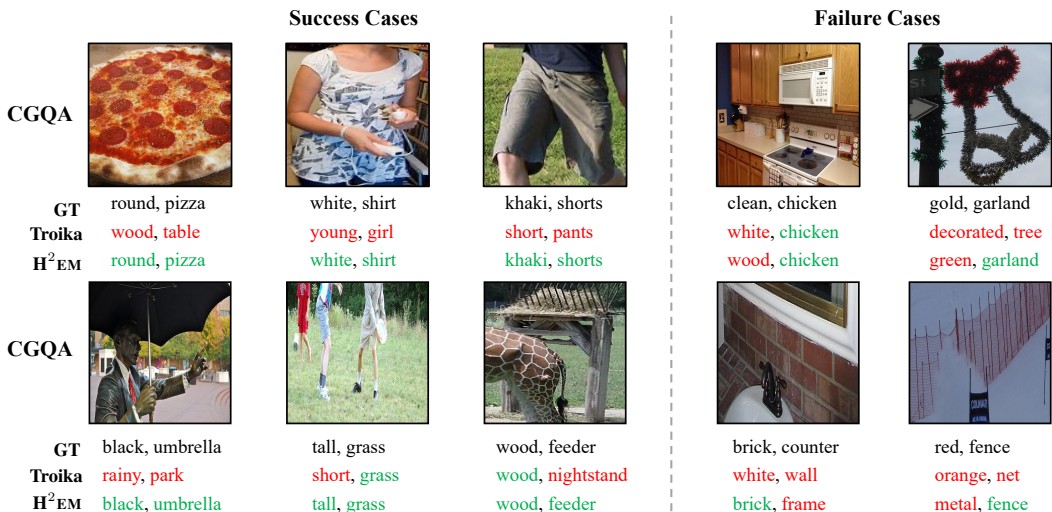

Figure S2: Visualization of Top-1 Predictions (§4.4). The comparison between our H²EM and Troika (Huang et al., 2024) on CGQA (Naeem et al., 2021). Correct and wrong predictions are marked in **green** and **red**, respectively.

strengths. Even when incorrect, its predictions are often more semantically grounded. In several examples, H²EM correctly identifies the object `garland`, `fence`, and predicts a wrong but still reasonable state, such as `wood` and `green`, while Troika misidentifies the object entirely. These visualizations further prove that our hierarchically structured space enables more robust predictions, even when faced with complex visual data.

## D.2 VISUALIZATION OF HYPERBOLIC SPACE

A key attribute of our framework is its ability to instill a consistent hierarchical geometry among both textual and visual modalities. To validate this for the visual domain, we visualized the learned hyperbolic space for visual embeddings from MIT-States in Figure S3. We used the same samples with text embeddings, *i.e.*, 2K for norm distribution (Desai et al., 2023) and 200 for HoroPCA (Chami et al., 2021). The hierarchical organization

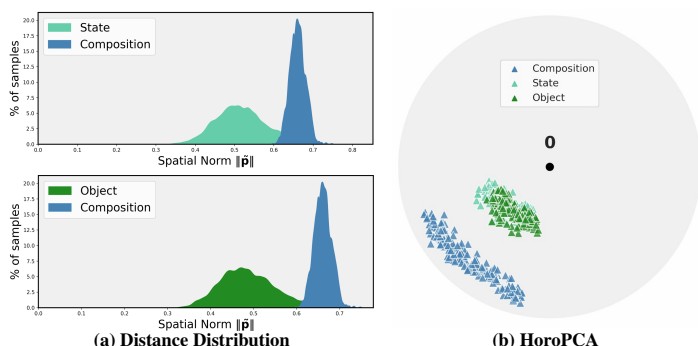

Figure S3: Visual results (§D) of visual embeddings in learned hyperbolic space using samples from MIT-States (Isola et al., 2015).

of the visual space is first demonstrated quantitatively by the norm distributions in Figure S3a. Here, embeddings for primitives (state and object) are clearly concentrated at a smaller radius from the origin than the more specific composition embeddings. This geometric arrangement is then visually confirmed in the HoroPCA projection (*cf.* Figure S3b), which shows primitive features forming central clusters while compositional features populate the periphery. This multi-modal consistency further validates our approach, confirming that our geometric objectives can structure the hierarchy.

## E REPRODUCIBILITY STATEMENT

We are committed to ensuring the reproducibility of our work. The complete source code for our H²EM framework, including model implementation, training scripts, evaluation protocols, and checkpoints, will be made publicly available upon publication. All experiments were conducted

on three publicly available benchmarks: MIT-States (Isola et al., 2015), UT-Zappos (Naeem et al., 2021), and CGQA (Yu & Grauman, 2014), with detailed descriptions of the datasets and their standard splits provided in §4.1. The implementation details of our model architecture, training procedures, and hyperparameters are provided in Appendix §A. Furthermore, the exact prompt used to automatically construct the semantic hierarchy is also included in the Appendix §C.

## F  LARGE LANGUAGE MODEL USAGE STATEMENT

We utilized the Gemini-2.5 Pro (Comanici et al., 2025) as LLM in two main capacities for this work. Primarily, it was used as a writing assistant to polish the language of the manuscript. Additionally, it is utilized to automatically construct the semantic hierarchy, as detailed in Appendix §C.

