# OpenReview forum: "Learning Hierarchical Hyperbolic Embeddings for Compositional Zero-Shot Learning"
_ICLR.cc/2026/Conference — ICLR 2026 Conference Withdrawn Submission_

### Official Review · Reviewer_PmLA · 2025-10-28

**Soundness:** 2
**Presentation:** 3
**Contribution:** 2
**Rating:** 2
**Confidence:** 3

**Summary:**

The authors present a framework for hyperbolic zero-shot learning using the hyperbolic geometry to better preserve the semantic and compositional hierarchies of the data without distortion. They present a taxonomic entailment loss to enforce structural relationships of the desired hierarchy. Additionally, a discriminative loss with hard negative mining is introduced to help improve fine-grained discrimination. The method presents strong empirical performance improvements over euclidean baselines. However, there are some missing acknowledgements of prior works that introduce similar hyperbolic terms and comparisons against prior hyperbolic zero-shot methods are omitted.

**Strengths:**

- The rationale behind the hyperbolic geometry for hierarchical representation learning is sensible and growing research field. The authors present the background and preliminaries well to justify the choice of geometry and explain the problem setting.
- The work maintains the hierarchical structure with the use of parent child entailment losses, the hard-negative mining is an again a useful addition to ensure non-uniform sampling of contrastive negatives.
- The results presented show clear performance benefit of employing the hyperbolic space for this problem domain. Furthermore, ablations are presented outlining the loss terms, and sensitivity analysis provided for the weighing of each loss terms.

**Weaknesses:**

**Major:**
- Two challenges are identified including hierarchical collapse, and that the contrastive objective is insufficient for fine-grained discrimination. However, no evidence, empirical or theoretical is provided to prove this and hence justify the rationale for the work.
- The proposed method is not necessarily novel, the taxonomic entailment loss has previously been introduced in a number of other works. Notably the definition of the taxonomic entailment loss is identical to that proposed in Pal, et al., (2025) Compositional Entailment Learning for Hyperbolic Vision-Language Model. The authors had not appropriately referenced the work in section 3.2.3.
- Some hyperbolic zero-shot learning methods, albeit from 2020 are not compared against here: Liu, et al., (2020) Hyperbolic Visual Embedding Learning for Zero-Shot Recognition.
- The metrics used to evaluate performance do not evaluate the hierarchies that the authors propose to capture. It is assumed that the hyperbolic space will capture such hierarchies but some analysis of the learning representation space to confirm the hierarchical structure would be beneficial. While the visualisation is presented a quantitative metric should be employed to confirm and compare.
    - Figure 4 is mostly uninformative given these are cherry-picked results.

**Minor:**
- There are some missing related works that utilise the entailment concepts presented in this work: Wang, et al., (2025) Learning Visual Hierarchies in Hyperbolic Space for Image Retrieval
- The introduction has some repeated text regarding the contributions, where the authors explain their losses. While it is introduced for clarity it is redundant.
- Much of the hyperbolic preliminaries that make up a significant proportion of the manuscript could be omitted to the supplementary, these are contributions of other works and foundations of the approach presented here. While nice to have in the manuscript, 1.5 to 2 pages are given to introducing these which is perhaps too much.
- The addition of code for reproducibility and validation during review would have been beneficial.
- While I respect the computational costs required for such experimentation, the error over multiple runs would be a nice addition to see how significant some of the smaller improvements are over baselines.

**Questions:**

1. How does the proposed loss differ from Pal, et al., (2025) Compositional Entailment Learning for Hyperbolic Vision-Language Model?
2. Have you compared to other hyperbolic methods?
3. Given the curvature was learnable, what was the resulting final curvatures for the model on each dataset?
4. What are the limitations of the method presented?

---

### Official Review · Reviewer_TtbD · 2025-10-29

**Soundness:** 3
**Presentation:** 3
**Contribution:** 3
**Rating:** 6
**Confidence:** 4

**Summary:**

This paper addresses CZSL via a new hyperbolic embedding framework H^2EM, capable of representing the rich semantic and conceptual hierarchy in compositional classes. Besides, it designs two loss objectives specially tailored for CZSL: a taxonomic entailment loss to structure symbolic hierarchy, and a discriminative alignment loss to separate fine-grained compositions. Extensive experiments on both closed-world and open-world setups demonstrate consistent improvements against prior state-of-the-arts.

**Strengths:**

This paper is well written and organized.
The motivation is clear, and the key components should be suited for tackling the diffculites.
The compared results are comprehensive including extensive ablation studies.

**Weaknesses:**

The novelty in this work mainly relies on two new loss functions, with a lack of new insights on empirical theory and network structures.

The hard negative construction is very common, and has been widely used in prior works (also known as semi-negatives). It looks not very related to hyperbolic space.

**Questions:**

In Eq.(16), it is unclear about why it defines three loss weights. In general, one weight can be set to 1 by default. In addition, in Sec. 4.3, when β1 = 1.0, the results are best, which proves the redundancy.

In Figure 4, why the proposed method is compared to Troika instead of other more competitive ones like LOGICZSL and PLO.

---

### Official Review · Reviewer_DNUC · 2025-10-30

**Soundness:** 2
**Presentation:** 3
**Contribution:** 2
**Rating:** 2
**Confidence:** 4

**Summary:**

This paper proposes a framework called H2EM, designed to learn hierarchical hyperbolic embeddings for Compositional Zero-Shot Learning (CZSL). The authors note that existing works usually optimize in Euclidean space, which cannot effectively capture the semantic hierarchy and conceptual structure inherent in compositional concepts. To address this, the paper introduces hyperbolic geometry modeling, leveraging its exponential volume growth property to embed tree-like hierarchical structures.
The core of the method includes two new loss functions:
1.	Taxonomic Entailment Loss — ensures hierarchical constraints between parent and child concepts;
2.	Discriminative Alignment Loss — incorporates hard negative mining to improve fine-grained discriminative ability.
On three CZSL datasets — MIT-States, UT-Zappos, and C-GQA — H2EM achieves new state-of-the-art performance in both closed-world and open-world settings.

**Strengths:**

1. The idea of incorporating hyperbolic space into CZSL is relatively new. The authors analyze the limitations of Euclidean spaces in representing large-scale hierarchical embeddings and provide a theoretically motivated alternative.
2. The paper conducts extensive comparisons against over ten strong baselines (including LOGICZSL, Troika, PLID, etc.) and provides ablation studies verifying the effectiveness of each module, with consistent results across datasets.

**Weaknesses:**

1. Although the paper claims a geometrically novel framework, its core components (Lorentz model, entailment cone, InfoNCE loss, hard negative mining) are direct adaptations of prior work. The innovation lies more in combinational application than theoretical advancement.
2. The model contains multiple nested modules (HCA module, triple-branch feature extractor, hyperbolic mapping, two loss functions). Despite good empirical results, there is little interpretive analysis explaining how hyperbolic space concretely improves performance. Figure 5 only visualizes structural distribution without linking it to quantitative gains.
3. The paper is mathematically dense, making it challenging to follow. Key details such as gradient stability in loss computation and curvature parameter (κ) selection are not discussed.
4. The method is heavily tailored to CZSL, and it remains unclear whether it can generalize to other compositional reasoning or cross-modal tasks, limiting its practical impact.

**Questions:**

1.  Can the authors provide more comparative analysis to quantitatively demonstrate the semantic structure preservation advantage of hyperbolic embeddings over Euclidean ones?
2. Have the authors explored different or adaptive curvature settings? Fixing κ may limit representational capacity.
3. Could the approach be extended beyond visual-language embeddings to multi-modal tasks, such as action composition or relational reasoning?

---

### Official Review · Reviewer_tHAw · 2025-10-30

**Soundness:** 3
**Presentation:** 3
**Contribution:** 3
**Rating:** 4
**Confidence:** 5

**Summary:**

The paper proposes a framework that learns Hierarchical Hyperbolic EMbeddings for Compositional zero-shot learning. The framework consists of two key components: a taxonomic entailment loss to enforce predefined hierarchies and a discriminative alignment loss with hard negative mining to enhance fine-grained discrimination. Extensive experiments on three benchmarks demonstrate that H2EM achieves state-of-the-art performance in both closed-world and open-world settings.

**Strengths:**

1. The framework addresses the limitations of Euclidean space in capturing the rich hierarchical structures of compositional concepts by employing hierarchical hyperbolic embeddings.

2. The taxonomic entailment loss enforces geometric constraints to ensure that child concepts lie within the entailment cones of their more general parent concepts, thereby effectively preventing hierarchical collapse.

**Weaknesses:**

1. The paper could further discuss the computational complexity of hyperbolic geometry versus traditional Euclidean methods.

2. The paper's direct use of large language models to automatically generate semantic hierarchies may lack precision and could require human intervention to enhance accuracy.

3. The paper should provide more theoretical explanations on why hyperbolic space can "better accommodate hierarchical structures."

4. The paper is advised to offer a clearer explanation of the three-path paradigm in the sections where it is mentioned.

5. This work is very similar to ''Compositional entailment learning for hyperbolic vision-language models''  in ICLR2025. Please discuss the difference.

**Questions:**

See Weaknesses.

---

### Note · Authors · 2025-11-13

I have read and agree with the venue's withdrawal policy on behalf of myself and my co-authors.